# Scalable Methods for Nonnegative Matrix Factorizations of Near-separable Tall-and-skinny Matrices

**Austin R. Benson**
ICME
Stanford University
Stanford, CA
arbenson@stanford.edu

**Jason D. Lee**
ICME
Stanford University
Stanford, CA
jdl17@stanford.edu

**Bartek Rajwa**
Bindley Biosciences Center
Purdue University
West Lafeyette, IN
brajwa@purdue.edu

**David F. Gleich**
Computer Science Department
Purdue University
West Lafeyette, IN
dgleich@purdue.edu

## Abstract

Numerous algorithms are used for nonnegative matrix factorization under the assumption that the matrix is nearly separable. In this paper, we show how to make these algorithms scalable for data matrices that have many more rows than columns, so-called "tall-and-skinny matrices." One key component to these improved methods is an orthogonal matrix transformation that preserves the separability of the NMF problem. Our final methods need to read the data matrix only once and are suitable for streaming, multi-core, and MapReduce architectures. We demonstrate the efficacy of these algorithms on terabyte-sized matrices from scientific computing and bioinformatics.

## 1 Nonnegative matrix factorizations at scale

A nonnegative matrix factorization (NMF) for an $m \times n$ matrix $X$ with real-valued, nonnegative entries is

$$X = WH \tag{1}$$

where $W$ is $m \times r$, $H$ is $r \times n$, $r < \min(m, n)$, and both factors have nonnegative entries. While there are already standard dimension reduction techniques for general matrices such as the singular value decomposition, the advantage of NMF is in *interpretability* of the data. A common example is facial image decomposition [17]. If the columns of $X$ are pixels of a facial image, the columns of $W$ may be facial features such as eyes or ears, and the coefficients in $H$ represent the intensity of these features. For this reason, among a host of other reasons, NMF is used in a broad range of applications including graph clustering [21], protein sequence motif discovery [20], and hyperspectral unmixing [18].

An important property of matrices in these applications and other massive scientific data sets is that they have many more rows than columns ($m \gg n$). For example, this matrix structure is common in big data applications with hundreds of millions of samples and a small set of features—see, e.g., Section 4.2 for a bioinformatics application where the data matrix has 1.6 billion rows and 25 columns. We call matrices with many more rows than columns *tall-and-skinny*. The number of columns of these matrices is small, so there is no problem storing or manipulating them. Our use

of NMF is then to uncover the hidden structure in the data rather than for dimension reduction or compression.

In this paper, we present scalable and computationally efficient NMF algorithms for tall-and-skinny matrices as prior work has not taken advantage of this structure for large-scale factorizations. The advantages of our method are: we preserve the geometry of the problem, we only read the data matrix once, and we can test several different nonnegative ranks ($r$) with negligible cost. Furthermore, we show that these methods can be implemented in parallel (Section 3) to handle large data sets. In Section 2.3, we present a new dimension reduction technique using orthogonal transformations. These transformations are particularly effective for tall-and-skinny matrices and lead to algorithms that only need to read the data matrix once. We compare this method with a Gaussian projection technique from the hyperspectral unmixing community [5, 7]. We test our algorithms on data sets from two scientific applications, heat transfer simulations and flow cytometry, in Section 4. Our new dimension reduction technique outperforms Gaussian projections on these data sets. In the remainder of the introduction, we review the state of the art for computing non-negative matrix factorizations.

## 1.1 Separable NMF

We first turn to the issue of how to practically compute the factorization in Equation (1). Unfortunately, for a fixed non-negative rank $r$, finding the factors $W$ and $H$ for which the residual $\|X - WH\|$ is minimized is NP-complete [26]. To make the problem tractable, we make assumptions about the data. In particular, we require a separability condition on the matrix. A nonnegative matrix $X$ is *separable* if

$$X = X(:, \mathcal{K})H,$$

where $\mathcal{K}$ is an index set with $|\mathcal{K}| = r$ and $X(:, \mathcal{K})$ is Matlab notation for the matrix $X$ restricted to the columns indexed by $\mathcal{K}$. Since the coefficients of $H$ are nonnegative, all columns of $X$ live in the conical hull of the "extreme" columns indexed by $\mathcal{K}$. The idea of separability was developed by Donoho and Stodden [15], and recent work has produced tractable NMF algorithms by assuming that $X$ almost satisfies a separability condition [3, 6].

A matrix $X$ is *noisy r-separable* or *near-separable* if $X = X(:, \mathcal{K})H + N$, where $N$ is a noise matrix whose entries are small. Near-separability means that all data points approximately live in the conical hull of the extreme columns. The algorithms for near-separable NMF are typically based on convex geometry (see Section 2.1) and can be described by the same two-step approach:

1. Determine the extreme columns, indexed by $\mathcal{K}$, and let $W = X(:, \mathcal{K})$.
2. Solve $H = \arg\min_{Y \in \mathbb{R}_+^{r \times n}} \|X - WY\|$.

The bulk of the literature is focused on the first step. In Section 3, we show how to implement both steps in a single pass over the data and provide the details of a MapReduce implementation. We note that separability (or near-separability) is a severe and restrictive assumption. The tradeoff is that our algorithms are extremely scalable and provably correct under this assumption. In big data applications, scalability is at a premium, and this provides some justification for using separability as a tool for exploratory data analysis. Furthermore, our experiments on real scientific data sets in Section 4 under the separability assumption lead to new insights.

## 1.2 Alternative NMF algorithms and related work

There are several approaches to solving Equation (1) that do not assume the separability condition. These algorithms typically employ block coordinate descent, optimizing over $W$ and $H$ while keeping one factor fixed. Examples include the seminal work by Lee and Seung [23], alternating least squares [10], and fast projection-based least squares [19]. Some of these methods are used in MapReduce architectures at scale [24].

Alternating methods require updating the entire factor $W$ or $H$ after each optimization step. When one of the factors is large, repeated updates can be prohibitively expensive. The problem is exacerbated in Hadoop MapReduce, where intermediate results are written to disk. In addition, alternating methods can take an intolerable number of iterations to converge. Regardless of the approach or computing platform, the algorithms are too slow when the matrices cannot fit in main memory In

contrast, we show in Sections 2 and 3 that the separability assumption leads to algorithms that do not require updates to large matrices. This approach is scalable for large tall-and-skinny matrices in big data problems.

## 2 Algorithms and dimension reduction for near-separable NMF

There are several popular algorithms for near-separable NMF, and they are motivated by convex geometry. The goal of this section is to show that when $X$ is tall-and-skinny, we can apply dimension reduction techniques so that established algorithms can execute on $n \times n$ matrices, rather than the original $m \times n$. Our new dimension reduction technique in Section 2.3 is also motivated by convex geometry. In Section 3, we leverage the dimension reduction into scalable algorithms.

### 2.1 Geometric algorithms

There are two geometric strategies typically employed for near-separable NMF. The first deals with conical hulls. A *cone* $C \subset \mathbb{R}^m$ is a non-empty convex set with $C = \{\sum_i \alpha_i x_i \mid \alpha_i \in \mathbb{R}_+, x_i \in \mathbb{R}^m\}$. The $x_i$ are generating vectors. In separable NMF,

$$X = X(:, \mathcal{K})H$$

implies that all columns of $X$ lie in the cone generated by the columns indexed by $\mathcal{K}$. For any $k \in \mathcal{K}$, $\{\alpha X(:, k) \mid \alpha \in \mathbb{R}_+\}$ is an *extreme ray* of this cone, In other words, the set of columns indexed by $\mathcal{K}$ are the set of extreme rays of the cone. The goal of the XRAY algorithm [22] is to find these extreme rays (i.e., to find $\mathcal{K}$). In particular, the greedy variant of XRAY selects the maximum column norm $\arg \max_j \|R^T X(:, j)\|_2 / \|X(:, j)\|_2$, where $R$ is a residual matrix that gets updated with each new extreme column.

The second approach deals with convex hulls, where the columns of $X$ are $\ell_1$-normalized. If $D$ is a diagonal matrix with $D_{ii} = \|X(:, i)\|_1$ and $X$ is separable, then

$$XD^{-1} = X(:, \mathcal{K})D(\mathcal{K}, \mathcal{K})^{-1}D(\mathcal{K}, \mathcal{K})HD^{-1} = (XD^{-1})(:, \mathcal{K})\tilde{H}.$$

Thus, $XD^{-1}$ is also separable (in fact, this holds for any nonsingular diagonal matrix $D$). Since the columns are $\ell_1$-normalized, the columns of $\tilde{H}$ have non-negative entries and sum to one. In other words, all columns of $XD^{-1}$ are in the convex hull of the columns indexed by $\mathcal{K}$. The problem of determining $\mathcal{K}$ is reduced to finding the extreme points of a convex hull. Popular approaches in the context of NMF include the Successive Projection Algorithm (SPA, [2]) and its generalization [16]. Another alternative, based on linear programming, is Hott Topixx [6]. Other geometric approaches had good heuristic performance [9, 25] before the more recent theoretical work. As an example of the particulars of one such method, SPA, which we will use in Section 4, finds extreme points by computing $\arg \max_j \|R(:, j)\|_2^2$, where $R$ is a residual matrix related to the data matrix $X$.

In any algorithm, we call the columns indexed by $\mathcal{K}$ *extreme columns*. The next two subsections are devoted to dimension reduction techniques for finding the extreme columns in the case when $X$ is tall-and-skinny.

### 2.2 Gaussian projection

A common dimension reduction technique is random Gaussian projections, and the idea has been used in hyperspectral unmixing problems [5]. In the hyperspectral unmixing literature, the separability is referred to as the *pure-pixel assumption*, and the random projections are motivated by convex geometry [7]. In particular, given a matrix $G \in \mathbb{R}^{m \times k}$ with Gaussian i.i.d. entries, the extreme columns of $X$ are taken as the extreme columns of $G^T X$, which is of dimension $k \times n$. Recent work shows that when $X$ is nearly $r$-separable and $k = O(r \log r)$, then all of the extreme columns are found with high probability [13].

### 2.3 Orthogonal transformations

Our new alternative dimension reduction technique is also motivated by convex geometry. Consider a cone $C \subset \mathbb{R}^m$ and a nonsingular $m \times m$ matrix $M$. It is easily shown that $x$ is an extreme ray of $C$

if and only if $Mx$ is an extreme ray of $MC = \{Mz \mid z \in C\}$. Similarly, for any convex set, invertible transformations preserve extreme points.

We take advantage of these facts by applying specific orthogonal transformations as the nonsingular matrix $M$. Let $X = Q\tilde{R}$ and $X = U\tilde{\Sigma}V^T$ be the *full* QR factorization and singular value decomposition (SVD) of $X$, so that $Q$ and $U$ are $m \times m$ orthogonal (and hence nonsingular) matrices. Then

$$Q^T X = \begin{pmatrix} R \\ \mathbf{0} \end{pmatrix}, \quad U^T X = \begin{pmatrix} \Sigma V^T \\ \mathbf{0} \end{pmatrix},$$

where $R$ and $\Sigma$ are the top $n \times n$ blocks of $\tilde{R}$ and $\tilde{\Sigma}$ and $\mathbf{0}$ is an $(m - n) \times n$ matrix of zeroes. The zero rows provide no information on which columns of $Q^T X$ or $U^T X$ are extreme rays or extreme points. Thus, we can restrict ourselves to finding the extreme columns of $R$ and $\Sigma V^T$. These matrices are $n \times n$, and we have significantly reduced the dimension of the problem. In fact, if $X = X(:, \mathcal{K})H$ is a separable representation, we immediately have separated representations for $R$ and $\Sigma V^T$:

$$R = R(:, \mathcal{K})H, \quad \Sigma V^T = \Sigma V^T(:, \mathcal{K})H.$$

We note that, although any invertible transformation preserves extreme columns, many transformations will destroy the geometric structure of the data. However, orthogonal transformations are either rotations or reflections, and they preserve the data's geometry. Also, although $Q^T$ and $U^T$ are $m \times m$, we will only apply them *implicitly* (see Section 3.1), i.e., these matrices are never formed or computed.

This dimension reduction technique is exact when $X$ is $r$-separable, and the results will be the same for orthogonal transformations $Q^T$ and $U^T$. This is a consequence of the transformed data having the same separability as the original data. The SPA and XRAY algorithms briefly described in Section 2.1 only depend on computing column 2-norms, which are preserved under orthogonal transformations. For these algorithms, applying $Q^T$ or $U^T$ preserves the column 2-norms of the data, and the selected extreme columns are the same. However, other NMF algorithms do not possess this invariance. For this reason, we present both of the orthogonal transformations.

Finally, we highlight an important benefit of this dimension reduction technique. In many applications, the data is noisy and the separation rank ($r$ in Equation (1)) is not known *a priori*. In Section 2.4, we show that the $H$ factor can be computed in the small dimension. Thus, it is viable to try several different values of the separation rank and pick the best one. This idea is extremely useful for the applications presented in Section 4, where we do not have a good estimate of the separability of the data.

## 2.4   Computing H

Selecting the extreme columns indexed by $\mathcal{K}$ completes one half of the NMF factorization in Equation (1). How do we compute $H$? We want $H = \arg\min_{Y \in \mathbb{R}_+^{r \times n}} \|X - X(:, \mathcal{K})Y\|^2$ for some norm. Choosing the Frobenius norm results in a set of $n$ nonnegative least squares (NNLS) problems:

$$H(:, i) = \arg\min_{y \in \mathbb{R}_+^r} \|X(:, \mathcal{K})y - X(:, i)\|_2^2, \quad i = 1, \dots, n.$$

Let $X = Q\tilde{R}$ with $R$ the upper $n \times n$ block of $\tilde{R}$. Then $H(:, i)$ is computed by finding $y \in \mathbb{R}_+^r$ that minimizes

$$\|X(:, \mathcal{K})y - X(:, i)\|_2^2 = \|Q^T (X(:, \mathcal{K})y - X(:, i))\|_2^2 = \|R(:, \mathcal{K})y - R(:, i)\|_2^2$$

Thus, we can solve the NNLS problem with matrices of size $n \times n$. After computing just the small $R$ factor from the QR factorization, we can compute the entire nonnegative matrix factorization by working with matrices of size $n \times n$. Analogous results hold for the SVD, where we replace $Q$ by $U$, the left singular vectors. In Section 3, we show that these computations are simple and scalable. Since $m \gg n$, computations on $O(n^2)$ data are fast, even in serial. Finally, note that we can also compute the residual in this reduced space, i.e.:

$$\min_{y \in \mathbb{R}_+^n} \|X(:, \mathcal{K})y - X(:, i)\|_2^2 = \min_{y \in \mathbb{R}_+^n} \|R(:, \mathcal{K})y - R(:, i)\|_2^2.$$

This simple fact is significant in practice. When there are several candidate sets of extreme columns $\mathcal{K}$, the residual error for each set can be computed quickly. In Section 4, we compute many residual errors for different sets $\mathcal{K}$ in order to choose an optimal separation rank.

We have now shown how to use dimension reduction techniques for tall-and-skinny matrix data in near-separable NMF algorithms. Following the same strategy as many NMF algorithms, we first compute extreme columns and then solve for the coefficient matrix $H$. Fortunately, once the upfront cost of the orthogonal transformation is complete, both steps can be computed using $O(n^2)$ data.

## 3   Implementation

Remarkably, when the matrix is tall-and-skinny, we only need to read the data matrix once. The reads can be performed in parallel, and computing platforms such as MapReduce, Spark, distributed memory MPI, and GPUs can all achieve optimal parallel communication. For our implementation, we use Hadoop MapReduce for convenience.[1] While all of the algorithms use sophisticated computation, these routines are only ever invoked with matrices of size $n \times n$. Furthermore, the local memory requirements of these algorithms are only $O(n^2)$. Thus, we get extremely scalable implementations. We note that, using MapReduce, computing $G^T X$ for the Gaussian projection technique is a simple variation of standard methods to compute $X^T X$ [4].

### 3.1   TSQR and R-SVD

The thin QR factorization of an $m \times n$ real-valued matrix $X$ with $m > n$ is $X = QR$ where $Q$ is an $m \times n$ orthogonal matrix and $R$ is an $n \times n$ upper triangular matrix. This is precisely the factorization we need in Section 2. For our purposes, $Q^T$ is applied implicitly, and we only need to compute $R$. When $m \gg n$, communication-optimal algorithms for computing the factorization are referred to as TSQR [14]. Implementations and specializations of the TSQR ideas are available in several environments, including MapReduce [4, 11], distributed memory MPI [14], and GPUs [1]. All of these methods avoid computing $X^T X$ and hence are numerically stable.

The thin SVD used in Section 2.3 is a small extension of the thin QR factorization. The thin SVD is $X = U \Sigma V^T$, where $U$ is $m \times n$ and orthogonal, $\Sigma$ is diagonal with decreasing, nonnegative diagonal entries, and $V$ is $n \times n$ and orthogonal. Let $X = QR$ be the thin QR factorization of $X$ and $R = U_R \Sigma V^T$ be the SVD of $R$. Then $X = (QU_R)\Sigma V^T = U\Sigma V^T$. The matrix $U = QU_R$ is $m \times n$ and orthogonal, so this is the thin SVD of $X$. The dimension of $R$ is $n \times n$, so computing its SVD takes $O(n^3)$ floating point operations (flops), a trivial cost when $n$ is small. When $m \gg n$, this method for computing the SVD is called the $R$-SVD [8]. Both TSQR and $R$-SVD require $O(mn^2)$ flops. However, the dominant cost is data I/O, and TSQR only reads the data matrix once.

### 3.2   Column normalization

The convex hull algorithms from Section 2.1 and the Gaussian projection algorithm from Section 2.2 require the columns of the data matrix $X$ to be normalized. A naive implementation of the column normalization in a MapReduce environment is: (1) read $X$ and compute the column norms; (2) read $X$, normalize the columns, and write the normalized data to disk; (3) use TSQR on the normalized matrix. This requires reading the data matrix twice and writing $O(mn)$ data to disk once just to normalize the columns. The better approach is a single step: use TSQR on the unnormalized data $X$ and simultaneously compute the column norms. If $D$ is the diagonal matrix of column norms, then

$$X = QR \rightarrow XD^{-1} = Q(RD^{-1}).$$

The matrix $\hat{R} = RD^{-1}$ is upper triangular, so $Q\hat{R}$ is the thin QR factorization of the column-normalized data. This approach reads the data once and only writes $O(n^2)$ data. The same idea applies to Gaussian projection since $G^T(XD^{-1}) = (G^T X)D^{-1}$. Thus, our algorithms only need to read the data matrix once in all cases. (We refer to the algorithm output as selecting the columns and computing the matrix H, which is typically what is used in practice. Retrieving the entries from the columns of A from K does require a subsequent pass.)

## 4   Applications

In this section, we test our dimension reduction technique on massive scientific data sets. The data are nonnegative, but we do not know *a priori* that the data is separable. Experiments on synthetic

data sets are provided in an online version of this paper and show that our algorithms are effective and correct on near-separable data sets.[2]

All experiments were conducted on a 10-node, 40-core MapReduce cluster. Each node has 6 2-TB disks, 24 GB of RAM, and a single Intel Core i7-960 3.2 GHz processor. They are connected via Gigabit ethernet. We test the following three algorithms: (1) dimension reduction with the SVD followed by SPA; (2) Dimension reduction with the SVD followed by the greedy variant of the XRAY algorithm; (3) Gaussian projection (GP) as described in Section 2.2. We note that the greedy variant of XRAY is not exact in the separable case but works well in practice [22].

Using our dimension reduction technique, all three algorithms require reading the data only once. The algorithms were selected to be a representative set of the approaches in the literature, and we will refer to the three algorithms as SPA, XRAY, and GP. As discussed in Section 2.3, the choice of QR or SVD does not matter for these algorithms (although it may matter for other NMF algorithms). Thus, we only consider the SVD transformation in the subsequent numerical experiments.

## 4.1 Heat transfer simulation

The heat transfer simulation data contains the simulated heat in a high-conductivity stainless steel block with a low-conductivity foam bubble inserted in the block [12].[3] Each column of the matrix corresponds to simulation results for a foam bubble of a different radius. Several simulations for random foam bubble locations are included in a column. Each row corresponds to a three-dimensional spatial coordinate, a time step, and a bubble location. An entry of the matrix is the temperature of the block at a single spatial location, time step, bubble location, and bubble radius. The matrix is constructed such that columns near 64 have far more variability in the data – this is then responsible for additional "rank-like" structure. Thus, we would intuitively expect the NMF algorithms to select additional columns closer to the end of the matrix. (And indeed, this is what we will see shortly.) In total, the matrix has approximately 4.9 billion rows and 64 columns and occupies a little more than 2 TB on the Hadoop Distributed File System (HDFS).

The left plot of Figure 1 shows the relative error for varying separation ranks. The relative error is defined as $\|X - X(:, \mathcal{K})H\|_F^2 / \|X\|_F^2$. Even a small separation rank ($r = 4$) results in a small residual. SPA has the smallest residuals, and XRAY and GP are comparable. An advantage of our projection method is that we can quickly test many values of $r$. For the heat transfer simulation data, we choose $r = 10$ for further experiments. This value is near an "elbow" in the residual plot for the GP curve. We note that the original SPA and XRAY algorithms would achieve the same reconstruction error if applied to the entire data set. Our dimension reduction technique allows us to accelerate these established methods for this large problem.

The middle plot of Figure 1 shows the columns selected by each algorithm. Columns 5 through 30 are not extreme in any algorithm. Both SPA and GP select at least one column in indices one through four. Columns 41 through 64 have the highest density of extreme columns for all algorithms. Although the extreme columns are different for the algorithms, the coefficient matrix $H$ exhibits remarkably similar characteristics in all cases. Figure 2 visualizes the matrix $H$ for each algorithm. Each non-extreme column is expressed as a conic combination of only two extreme columns. In general, the two extreme columns corresponding to column $i$ are $j_1 = \arg\max\{j \in \mathcal{K} \mid j < i\}$ and $\arg\min\{j \in \mathcal{K} \mid j > i\}$. In other words, a non-extreme column is a conic combination of the two extreme columns that "sandwich" it in the data matrix. Furthermore, when the index $i$ is closer to $j_1$, the coefficient for $j_1$ is larger and the coefficient for $j_2$ is smaller. This phenomenon is illustrated in the right plot of Figure 1.

## 4.2 Flow cytometry

The flow cytometry (FC) data represent abundances of fluorescent molecules labeling antibodies that bind to specific targets on the surface of blood cells.[4] The phenotype and function of individual cells can be identified by decoding these label combinations. The analyzed data set contains measurements of 40,000 single cells. The measurement fluorescence intensity conveying the abundance

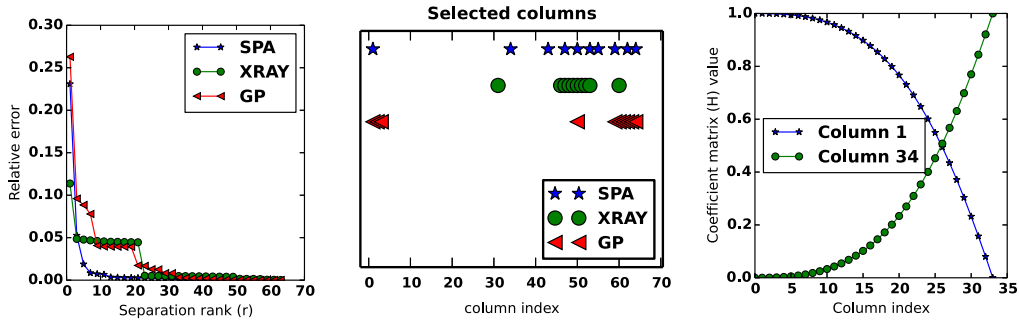

Figure 1: (Left) Relative error in the separable factorization as a function of separation rank ($r$) for the heat transfer simulation data. Our dimension reduction technique lets us test all values of $r$ quickly. (Middle) The first 10 extreme columns selected by SPA, XRAY, and GP. We choose 10 columns as there is an "elbow" in the GP curve there (left plot). The columns with larger indices are more extreme, but the algorithms still select different columns. (Right) Values of $H(\mathcal{K}^{-1}(1), j)$ and $H(\mathcal{K}^{-1}(34), j)$ computed by SPA for $j = 2, \ldots, 33$, where $\mathcal{K}^{-1}(1)$ and $\mathcal{K}^{-1}(34)$ are the indices of the extreme columns 1 and 34 in $W$ ($X = WH$). Columns 2 through 33 of $X$ are roughly convex combinations of columns 1 and 34, and are not selected as extreme columns by SPA. As $j$ increases, $H(\mathcal{K}^{-1}(1), j)$ decreases and $H(\mathcal{K}^{-1}(34), j)$ increases.

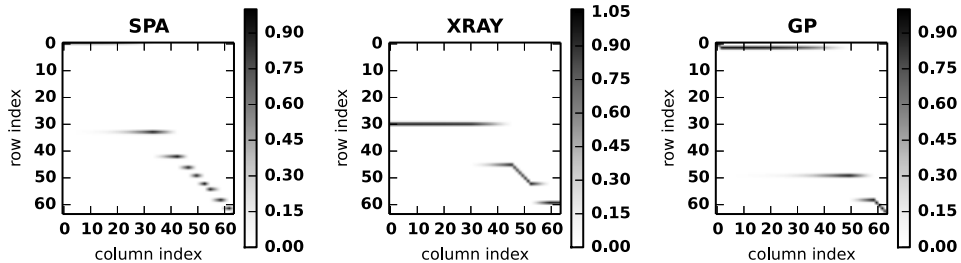

Figure 2: Coefficient matrix $H$ for SPA, XRAY, and GP for the heat transfer simulation data when $r = 10$. In all cases, the non-extreme columns are conic combinations of two of the selected columns, i.e., each column in $H$ has at most two non-zero values. Specifically, the non-extreme columns are conic combinations of the two extreme columns that "sandwich" them in the matrix. See the right plot of Figure 1 for a closer look at the coefficients.

information were collected at five different bands corresponding to the FITC, PE, ECD, PC5, and PC7 fluorescent labels tagging antibodies against CD4, CD8, CD19, CD45, and CD3 epitopes.

The measurements are represented as the data matrix $A$ of size $40,000 \times 5$. Our interest in the presented analysis was to study pairwise interactions in the data (cell vs. cell, and marker vs. marker). Thus, we are interested in the matrix $X = A \otimes A$, the Kronecker product of $A$ with itself. Each row of $X$ corresponds to a pair of cells and each column to a pair of marker abundance values. $X$ has dimension $40,000^2 \times 5^2$ and occupies 345 GB on HDFS.

The left plot of Figure 3 shows the residuals for the three algorithms applied to the FC data for varying values of the separation rank. In contrast to the heat transfer simulation data, the relative errors are quite large for small $r$. In fact, SPA has large relative error until nearly all columns are selected ($r = 22$). XRAY has the smallest residual for any value of $r$. The right plot of Figure 3 shows the columns selected when $r = 16$. XRAY and GP only disagree on one column. SPA chooses different columns, which is not surprising given the relative residual error. Interestingly, the columns involving the second marker defining the phenotype (columns 2, 6, 7, 8, 9, 10, 12, 17, 22) are underrepresented in all the choices. This suggests that the information provided by the second marker may be redundant. In biological terms, it may indicate that the phenotypes of the individual cells can be inferred from a smaller number of markers. Consequently, this opens a possibility that in modified experimental conditions, the FC researchers may omit this particular label, and still be able to recover the complete phenotypic information. Owing to the preliminary nature of these studies, a more in-depth analysis involving multiple similar blood samples would be desirable in order to confirm this hypothesis.

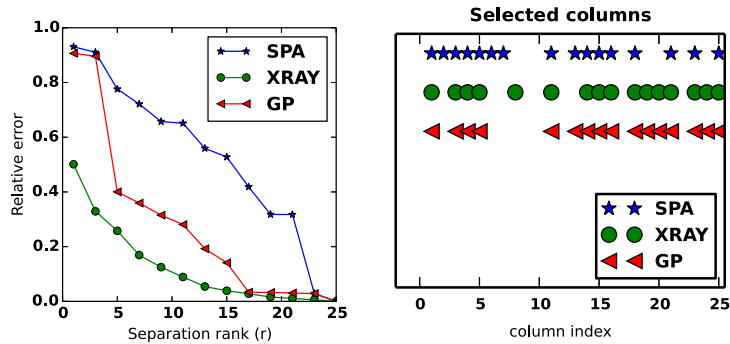

Figure 3: (Left) Relative error in the separable factorization as a function of nonnegative rank ($r$) for the flow cytometry data. (Right) The first 16 extreme columns selected by SPA, XRAY, and GP. We choose 16 columns since the XRAY and GP curve levels for larger $r$ (left plot).

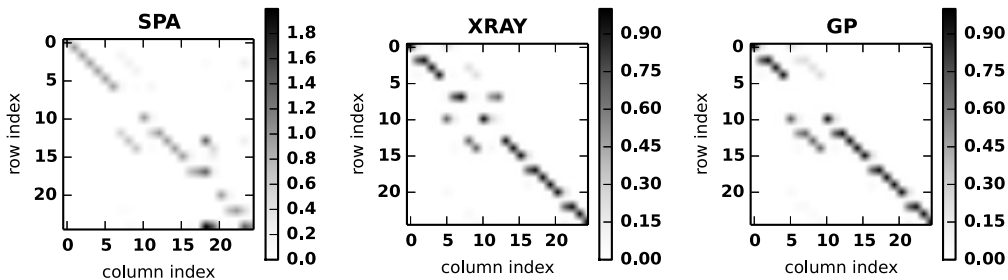

Figure 4: Coefficient matrix $H$ for SPA, XRAY, and GP for the flow cytometry data when $r = 16$. The coefficients tend to be clustered near the diagonal. This is remarkably different to the coefficients for the heat transfer simulation data in Figure 2.

Finally, Figure 4 shows the coefficient matrix $H$. The coefficients are larger on the diagonal, which means that the non-extreme columns are composed of nearby extreme columns in the matrix.

## 5 Discussion

We have shown how to compute nonnegative matrix factorizations at scale for near-separable tall-and-skinny matrices. Our main tool was TSQR, and our algorithms only needed to read the data matrix once. By reducing the dimension of the problem, we can easily compute the efficacy of factorizations for several values of the separation rank $r$. With these tools, we have computed the largest separable nonnegative matrix factorizations to date. Furthermore, our algorithms provide new insights into massive scientific data sets. The coefficient matrix $H$ exposed structure in the results of heat transfer simulations. Extreme column selection in flow cytometry showed that one of the labels used in measurements may be redundant. In future work, we would like to analyze additional large-scale scientific data sets. We also plan to test additional NMF algorithms.

The practical limits of our algorithm are imposed by the tall-and-skinny requirement where we assume that it is *easy* to manipulate $n \times n$ matrices. The synthetic examples we explored used up to 200 columns, and regimes up to 5000 columns have been explored in prior work [11]. A rough rule of thumb is that our implementations should be possible as long as an $n \times n$ matrix fits into main memory. This means that implementations based on our work will scale up to $30,000$ columns on machines with more than 8 GB of memory; although at this point communication begins to dominate. Solving these problems with more columns is a challenging opportunity for the future.

**Acknowledgments**

ARB and JDL are supported by an Office of Technology Licensing Stanford Graduate Fellowship. JDL is also supported by a NSF Graduate Research Fellowship. DFG is supported by NSF CAREER award CCF-1149756. BR is supported by NIH grant 1R21EB015707-01.

## Footnotes

[1]The code is available at https://github.com/arbenson/mrnmf.

[2]http://arxiv.org/abs/1402.6964.

[3]The heat transfer simulation data is available at https://www.opensciencedatacloud.org.

[4]The FC data is available at https://github.com/arbenson/mrnmf/tree/master/data.

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
