[Reviews · NeurIPS 2014]

Submitted by Assigned_Reviewer_1

This paper describes a non-negative matrix factorization for tall and skinny matrices. This algorithm works in the bigdata scenario because it need only pass over the tall skinny matrix one time.

This linear read of the tall skinny matrix does not fully utilize the distributed mapreduce framework. I wonder, is it possible to parallelize the reading of the matrix and combine the results from subsets of the data into one final result? Perhaps this is already being done, but is buried in the details of TQSR and not mentioned.

In figure 1, r=10 is chosen based on the GP curve, and then used to compare all 3 methods. A fairer approach would be to choose the "elbows" for each method independently. I'm also unclear on what the right hand subfigure of Figure 1 refers to. The text says the lines correspond to the rows, but the legend says they correspond to columns.

From a readability perspective, this paper could use some effort towards cohesion. Each section reads well, but it is harder to put the sections together to make a coherent story. For example, name your new extension and mention it early. Perhaps describe the content of the paper (i.e. a roadmap) in the introduction and remind the reader of where you are in each section.

If I understand correctly, you are applying your new technique to three existing algorithms. But there is no comparison (in either reconstruction error or time to compute) of your extension vs without your extension. For this reason it is unclear to me what the contribution of this paper really amounts to. You state that these results constitute the largest NMF to date, which is impressive, and so perhaps none of the 3 algorithms could be run on these data sets. If true, I suggest and exposition of run time on smaller simulated data sets to really illustrate the advantage your algorithm affords.

Minor comments:
line 253 simultaneous -> simultaneously

Summary: This paper performs non-negative matrix factorization on big data. The algorithm is well suited to tall and skinny matrices. The main contribution is a reformulation of the objective for 3 previously published NMF algorithms. Given that the contribution is an extension, I'm disappointed to see that no comparison to the original 3 algorithms is included.

Submitted by Assigned_Reviewer_19

This paper proposes scalable nonnegative matrix factorization (NMF) methods for the so-called “tall-and-skinny matrices” which have many more rows than columns. The basic idea is to adopt an orthogonal matrix transformation that preserves the separability of the NFM problem. Experiments on Terabyte-scale data sets are used to verify the effectiveness of the proposed methods.

The problem of scalable NMF is interesting and challenging. The proposed solution seems to be novel and effective for some special matrices. Experiments are also convincing.

One of the practical limits is that the proposed solution can only be used for “tall-and-skinny” separable matrices. However, if a matrix is both “tall-and-skinny” and separable, it might be not necessary to perform factorization in real applications.

Capitalize the first character of the words in the title.
Summary: The problem of scalable NMF is interesting and challenging. The proposed solution seems to be novel and effective for some special matrices. Experiments are also convincing.

One of the practical limits is that the proposed solution can only be used for “tall-and-skinny” separable matrices.

Submitted by Assigned_Reviewer_20

The paper presents a nonnegative matrix Factorization algorithm for tall and skinny matrices. The key insight eloitet is that orthogonal transformations, such as those obtained by SVD and QR Factorizations do preserve the needed structure to find the extreme set of the matrix. This allows a problem of size n x m to be transformed into one of size n x m, making it computationally tractable.

First of all, the paper is really well written and a pleasure to read.

I particularly liked how the paper fluently mixes theoretical concerns with practical ones. All of the theoretical steps are motivated by practical considerations. An example of such a consideration is MapReduce's inability to support iterations, which motivates the need for single pass algorithms.

I'd also like to highlight that the authors went to the trouble of making both their code and the data used available. This effort needs to be applauded.
Summary: Great paper that connects the dots between a theory insight and applications.
Author Feedback
Author rebuttal: We thank the reviewers for their careful reading of our manuscript, and we’d like to respond to a few specific concerns raised in reviews 1 and 2 regarding (I) the precise nature of our parallel algorithm, (II) our choice of experiments, (III) the overall utility of tall-and-skinny separable NMF, and finally, (IV) the presentation.

Concern I. We note that our algorithm can utilize a fully distributed read through the TSQR method as Reviewer 1 surmised. Prior work, notably [4] and [11], describe this method in detail. This makes the initial pass over the data (almost) fully parallelizable. The “sequential step” is on a small fraction of the total data (think of 1 in 10,000 rows, so a reduction in data by 4 order of magnitude).

Concern II. Review 1 raised questions about comparing our method with the original algorithms. We think of our method not as a new algorithm but rather as a way to enable current techniques to run on big data. We will get the same reconstruction errors as the original three algorithms. With regard to runtime scaling on “modest data” we present the following results that illustrate the value of our approach even on datasets that fit into L3 cache on a modern processor.

The matrix is 25,000-by-25.

original SPA: took 0.544147s
QR-first SPA: took 0.016151s

original XRAY: took 5.445988s
QR-first XRAY: took 0.043012s

The original methods are in-memory procedures and we lack the memory necessary to run them on our larger datasets to compare runtimes. We plan to add a few remarks to the paper to better elucidate our points here.

Concern III. Review 2 raised a concern about the limitations of tall-and-skinny data. We realize this is a limitation, but we do provide two applications. These applications are general in the sense that other scientific simulation data and flow cytometry data can be represented in this form. We are currently looking at business intelligence applications and bioinformatics applications with gene expression data. Also, randomized approximation techniques often reduce matrices to tall-and-skinny form, on which our methods can be employed.

One of the reasons we would be excited about presenting this work at a venue such as NIPS is that we anticipate there are other problems in machine learning and information processing that -- like these ones -- will benefit from a highly scalable separable non-negative matrix factorization that comes with reasonable performance guarantees and gives some guidance on how to “pick” the rank of the factorization.

Concern IV. Review 1 raised a few questions about some of our choices in exposition and presentation. We agree that we can make improvements here. Specifically, we can add more of a road map in the introduction and include reminders of how each section fits into the paper as a whole. Also we agree with Review 1 that we need to improve the description of Figure 1.

Here is a summary of what we want to convey with Figure 1. Our method allows us to compute curves like the left-hand-side plot on big data, and the domain scientist can pick a cut-off and reconstruction error for his or her particular application. The middle plot just shows that the three algorithms are actually picking out different columns. The right-hand-side plot is illustrating an interesting trend in the heat simulation data. As pointed out in Review 1, the description of this plot is not clear, and we will rectify that. The curves are the values of H(K(1), j) and H(K(34), j) for j = 2, …, 33. K(1) and K(34) are the indices of the extreme columns 1 and 34 in W (X = WH). Columns 2 through 33 of X are roughly convex combinations of columns 1 and 34. As j increases, H(K(1), j) decreases and H(K(34), j) increases.